# Dynamic Manipulation of THz Waves Enabled by Phase-Transition VO_2_ Thin Film

**DOI:** 10.3390/nano11010114

**Published:** 2021-01-06

**Authors:** Chang Lu, Qingjian Lu, Min Gao, Yuan Lin

**Affiliations:** 1School of Materials and Energy, University of Electronic Science and Technology of China, Chengdu 610054, China; luchang2010@foxmail.com (C.L.); luqingjian2013@163.com (Q.L.); 2State Key Laboratory of Electronic Thin Films and Integrated Devices, University of Electronic Science and Technology of China, Chengdu 610054, China; 3Medico-Engineering Cooperation on Applied Medicine Research Center, University of Electronic Science and Technology of China, Chengdu 610054, China

**Keywords:** vanadium dioxide, thin film, phase transition, external stimuli, active modulation, terahertz, metamaterials

## Abstract

The reversible and multi-stimuli responsive insulator-metal transition of VO_2_, which enables dynamic modulation over the terahertz (THz) regime, has attracted plenty of attention for its potential applications in versatile active THz devices. Moreover, the investigation into the growth mechanism of VO_2_ films has led to improved film processing, more capable modulation and enhanced device compatibility into diverse THz applications. THz devices with VO_2_ as the key components exhibit remarkable response to external stimuli, which is not only applicable in THz modulators but also in rewritable optical memories by virtue of the intrinsic hysteresis behaviour of VO_2_. Depending on the predesigned device structure, the insulator-metal transition (IMT) of VO_2_ component can be controlled through thermal, electrical or optical methods. Recent research has paid special attention to the ultrafast modulation phenomenon observed in the photoinduced IMT, enabled by an intense femtosecond laser (fs laser) which supports “quasi-simultaneous” IMT within 1 ps. This progress report reviews the current state of the field, focusing on the material nature that gives rise to the modulation-allowed IMT for THz applications. An overview is presented of numerous IMT stimuli approaches with special emphasis on the underlying physical mechanisms. Subsequently, active manipulation of THz waves through pure VO_2_ film and VO_2_ hybrid metamaterials is surveyed, highlighting that VO_2_ can provide active modulation for a wide variety of applications. Finally, the common characteristics and future development directions of VO_2_-based tuneable THz devices are discussed.

## 1. Introduction

The terahertz (THz) wave, which is defined as the electromagnetic spectrum (0.1–10 THz) between microwave radiation and infrared light, has attracted increasing attention since the 1980s [1,2,3]. Promoted by the femtosecond laser (fs laser), as well as the significantly improved THz generators and detectors, a couple of advanced THz technologies have been materialized, e.g., the well-matured terahertz time-domain spectroscopy (THz-TDs) that is capable of providing whole new insights into the material nature in the THz frequency range [4]. Vanadium dioxide (VO_2_), as one of the most important phase-change materials, was subsequently investigated to reveal the evolution of THz properties across its reversible first-order insulator-metal transition (IMT). Early reports in the 2000s have demonstrated that VO_2_ exhibits remarkable changes in the THz transmittance and reflectance in response to external thermal [5], optical [6,7] and electrical [8] stimuli. Such multi-stimuli responsive features, as well as the easily accessible transition temperature (341 K), make VO_2_ a promising material to fabricate dynamically tuneable THz devices [9]. More recently, the rapid development of multimedia service has caused an explosive demand for high-capacity wireless communications. THz communication technology has become increasingly important for the potential of increased bandwidth capacity compared to microwave systems [10,11,12,13,14]. The manipulation of the transmission properties of THz waves, such as amplitude, phase, polarization and spatiotemporal distribution, is based on the modulation effect of THz modulators, which is one of the core devices in the THz communication system. Practical applications require THz modulators capable of effectively manipulating the electromagnetic properties of THz waves and dynamically responding to external control signals, which significantly promote the research and application of VO_2_ in the THz regime [15,16,17,18,19,20,21,22]. Since the 2010s, extensive applications in THz regime based on VO_2_ have been demonstrated, such as amplitude modulators [23,24], tuneable absorbers [25], phase shifters [26], polarization converters [27,28,29], active frequency selective surfaces [30,31,32] and optical memory devices [33,34,35]. To date, VO_2_ has played an important role in THz devices as a phase-change material [36,37,38,39,40].

The IMT of VO_2_ has attracted extensive interest since it was observed by Morin in 1959 [41]. Generally, VO_2_ undergoes a reversible change in electric conductivity by several orders of magnitude at 341 K, accompanied by a simultaneous crystallographic phase transition (CPT) [42]. Despite the great efforts devoted to understanding the physical mechanism underlying the combined phase transition, debates still exist, largely due to the incapability of traditional thermal research in decoupling the IMT and CPT on a timescale [43,44,45,46,47,48]. Therefore, ultrafast pump-probe techniques have been widely used to give insights into the structural and electrical dynamics of VO_2_ in time [49]. This kind of research usually uses intense femtosecond pump laser to trigger the phase transition of VO_2_, while a delayed pulse of either THz radiation, X-rays or electrons is utilized to probe the evolution of IMT or CPT [50,51,52,53,54,55,56,57]. Thus, the phenomenon demonstrates that the IMT of VO_2_ can be completed within 1 ps while the CPT takes a relatively long time [57]. The ultrafast IMT triggered by the fs pump laser extensively broadens the applications based on VO_2_, making VO_2_ a promising candidate for high-speed THz modulators [58]. Except for the mentioned thermal and fs laser-based approach, recent efforts have shown that electrical field [59,60,61,62,63], continuous-wave (CW) laser [64,65,66], intense THz field [67,68,69] and electrochemical modification [70,71,72] can also provide effective control of IMT, and all these approaches can be integrated into THz devices.

Therefore, VO_2_ film is a natural multi-stimuli responsive THz modulator [9]. The modulation depth of transmission amplitude could reach up to 85% in high-quality epitaxial-grown VO_2_ films [73,74]. Additionally, since the transition in THz transmittance originates from the change of carrier density, the modulation phenomenon of VO_2_ films exhibits a broadband and nearly frequency-independent feature [75,76,77]. When combining VO_2_ film with subwavelength plasmonic structures, also called metamaterials [78,79,80,81,82,83], such as rectangular slot antennas [23,24,84], split-ring resonators [35] and grid lines [85], more complex functionalities can be realized. In such designs, VO_2_ is settled as the key component of the meta-atoms [86], such as resonators, dielectric layers and resonator gaps. When the IMT of VO_2_ is triggered by external stimuli, the plasmon spectrum of the VO_2_ hybrid meta-atoms will be changed, resulting in a transition in whole device response. Furthermore, since the dynamic control of the device is based on the IMT of VO_2_, memory effects originating from the intrinsic hysteresis behaviour of the first-order phase transition can be observed in these devices, presenting a potential for memory-type applications [33,34,35,72].

This review aims to provide a comprehensive survey of the recent advances in tuneable THz devices based on phase-change material VO_2_. The band theory and crystal structure, as well as the physical mechanism underlying the modulation phenomenon of VO_2_, are also introduced to understand the material nature. We focus special emphasis on the emerging ultrafast modulation approach enabled by the fs pump laser, as well as the unique memory phenomenon. Finally, the challenges and future perspectives of VO_2_-based active THz devices are considered.

## 2. VO_2_: Phase-Change Material

### 2.1. Crystal Structure & Band Structure

In traditional thermal studies, it is generally observed that VO_2_ undergoes reversible first-order IMT at a critical temperature (*T*_c_) of 341 K, accompanied by a remarkable modification of the crystallographic structure. As shown in Figure 1b, in high-temperature phase, VO_2_ exhibits a high symmetric rutile (R) structure—V cations occupy the centre site of oxygen octahedrons and equidistantly distribute along the rutile c axis with a V-V distance of 0.285 nm [44]. However, the symmetry breaks when the temperature is lowered to *T*_c_. The formed monoclinic phase is characterized by the formation of tilted V-V dimers, leading to the doubling of the unit cell, as shown in Figure 1a. The dimerization of V cations results in two different V–V distances, 0.265 nm (inside a dimer) and 0.312 nm (between dimers) [87]. Accompanying the structural transition, the band structure also changes, which is responsible for the remarkable transition in electronic conductivity. From the high temperature rutile phase to the low-temperature monoclinic phase, due to the formation of V-V dimers, the 3d_II_ band splits into two parts—the lower-energy, full-filled, bonding 3d_II_ band and the higher-energy, empty, antibonding 3d_II_* band, opening a bandgap of ~0.6 eV (see Figure 1c,d) [88].

As a classical phase-change material, VO_2_ has attracted considerable research interest over the years for its unique combined phase transition. However, since traditional thermal studies have difficulties in decoupling the IMT and CPT on a timescale, a long-standing debate over the underlying phase transition mechanism remains unsettled between two main alternative models—a lattice distortion-driven (Peirrls-like) transition or an electron correlation-driven (Mott-like) transition [46,49]. Hence, time-resolved ultrafast pump-probe techniques have been extensively used to detect the structural and electrical dynamics in the vicinity of the phase transition. Moreover, THz techniques have played an important role in this field, since the greatly improved time resolution of THz-TDs systems allows coherent investigation on electron dynamics on the timescale of femtoseconds [56,89]. Using THz radiation as a probe to detect the ultrafast electronic dynamics across the IMT is far superior to the conventional resistivity methods. Detailed introductions are given in Section 3.2.1.

### 2.2. Modulation Phenomenon in the THz Regime

The reorganization of band structure across the phase transition results in the release of free charge carriers, which is responsible for the modulation phenomenon in the THz transmittance. In order to investigate the mechanism underlying the modulation, THz-TDS measurements were carried out for VO_2_ thin films, and the resultant spectrum in frequency domain is shown in Figure 2a [90]. RA rmarkable decrease in THz transmission can be observed as the film is heated to metallic state. Other important characteristics of IMT, such as the reversibility and thermal hysteresis behaviour, can be demonstrated in Figure 2b, in which the evolution of THz transmission in the heating and cooling process is illustrated. For optical memory-type devices, a large hysteresis width is preferable to obtain stationary memory state, while a small hysteresis width is more suitable for applications that need fast erasure of the excited metallic state.

To further understand the modulation phenomenon of VO_2_ in the THz frequency range, two theoretical models, the Drude-Smith model and the Bruggeman EMT (effective medium theory, are introduced. These two models describe the THz conductivity of VO_2_ film from different perspectives. The former pays attention to the dependency of frequency, while the latter emphasizes the influence of the volume fraction of metallic phase. The Drude-Smith model has been extensively used to model the complex conductivity of VO_2_ films. This model is a classical generalization of the Drude model in order to involve the conductivity suppression effect caused by carrier localization [5,77,91,92].

The most common form is given by [77]:(1)$${\tilde{\sigma }}_{DS}\left(\omega \right)=\frac{ne^{2}\tau_{DS}/m^{*}}{1-i\omega \tau_{DS}}\left(1+\frac{c}{1-i\omega \tau_{DS}}\right)$$
where ${\tilde{\sigma }}_{DS}\left(\omega \right)$ is the complex conductivity, ω is the angular frequency, n is the electron density, $\tau_{DS}$ is the Drude–Smith scattering time, $m^{*}$ is the effective mass and c is a parameter that can vary between 0 (free Drude carriers) and −1 (fully localized carriers). In this formula, the parameters $\tau_{DS}$ and *c* contain localization details of carriers in the VO_2_ film and could be derived through fitting the measured complex terahertz conductivity data with the Drude-Smith formula.

Generally, the conductivity transition of single-domain VO_2_ crystals accompanied with the IMT is abrupt and step-like. However, for multidomain VO_2_ thin films, the conductivity transition is much more complex due to the dispersion of local phase-transition temperature in different domains [93,94,95]. The scanning infrared microscopy maps presented in Figure 2c directly demonstrate the coexistence of metallic and insulating domains in a nanostructured VO_2_ film [93]. As shown in Figure 2c, as the temperature increases, newly formed metallic domains initially nucleate, and then grow and connect until the entire film is in a metallic state. Therefore, the conductivity transition process that decides the modulation effect of the VO_2_ thin film has been widely described as a percolation process, in which the effective conductivity of the whole film can be described by the effective medium theory (EMT) [5,73,96]. The average conductivity of multiphase system modelled by EMT, which mainly concerns the volume fraction, depolarization factor and microscopic conductivity of different kinds of phases, is based on the general treatment of the electrostatic field around the inhomogeneous domains [97]. The most commonly used EMT formula in the VO_2_-related research is as follows [77]:(2)$$p\frac{\sigma_{m}-\sigma_{eff}}{g\sigma_{m}+\left(1-g\right)\sigma_{eff}}+\left(1-p\right)\frac{\sigma_{i}-\sigma_{eff}}{g\sigma_{i}+\left(1-g\right)\sigma_{eff}}=0$$
where p is the volume fraction of metallic domains and g is a shape-dependent parameter that governs the percolation threshold. $\sigma_{i}$, $\sigma_{m}$ and $\sigma_{eff}$ are the insulating-phase, metallic-phase and effective THz conductivities, respectively. Generally, the metallic phase volume fraction p is tuneable and highly responsive to external excitation strength, such as the temperature in thermal-excited IMT [73,98,99] or the laser fluence in photoexcited IMT [52,100].

Figure 3a offers a comparison of the two models, in which both of them are fitted to the complex conductivity of VO_2_ film at different temperatures [101]. The Drude-Smith model in Figure 3a fits the positive slope of the conductivity curve well, whereas the Bruggeman EMT model only fits the magnitude of the complex conductivity and has difficulties to describe the frequency-dependent changes. Such difference relies on the carrier localization effect that is involved in the Drude-Smith model but ignored in the EMT model. Despite the failure in frequency domain, the EMT model still plays an important role, since it establishes a relationship between the effective conductivity and stimuli strength through the phase fraction of metallic domains. As an example, as presented in Figure 3b, by fitting the representative conductivity points with EMT models, researchers can extract a general expression relating the complex THz conductivity to temperature [73].

In conclusion, the modulation phenomenon observed in VO_2_ film is characterized by the following features: Reversiblility, thermal hysteresis behaviour (memory effect), broad frequency band, high tuneability and responsiveness. Additionally, the critical temperature of VO_2_ is much closer to RT compared with other phase-change materials (PCMs) utilized for tuneable THz devices [40], such as superconductors [102,103,104,105], chalcogenides [106,107,108,109,110,111] and ferroelectrics [112,113,114,115], which means significant advantages in low energy consumption.

## 3. VO_2_: Multi-Stimuli Responsive Material

The IMT of VO_2_ can be triggered by diverse external stimuli [38], such as heating, photon, electric field [60], magnetic field, electrochemical modification [70] and mechanical strain. Controlling the IMT of VO_2_ through external stimuli is an active research area and related introductions can be found in several review articles [9,58]. However, not all of these methods can be utilized in the THz regime. For example, strain-induced IMT is usually carried out by introducing uniaxial compression strain along V-V chains of VO_2_ crystals. Such requirement can be satisfied by combing micro-actuators with one-dimensional single-crystal VO_2_ nanobeams. However, this requirement is difficult to realize in THz devices [116]. Here, we focus on the modulation approach that has been widely proved available in the THz regime, mainly including the thermal, optical and electrical methods. The underlying phase transition mechanisms are also presented in the following part to help understand the characteristics of different approaches.

### 3.1. Thermal-Excited IMT

The thermal approach is a fundamental method to control the phase transition of VO_2_. When temperature reaches 341 K, the IMT of VO_2_ will be triggered, accompanied by a simultaneous crystallographic transition. Thermodynamics study explains the driving force accounting for the combined phase transition as a competition between the higher entropy of the metallic phase, mainly provided by softer phonons, and the lower enthalpy of the insulating phase resulting from bandgap opening [49,117].

In practical applications, the temperature could be controlled either by discrete temperature controller or by an electrical-heating circuit integrated into the device [31,32,118]. The former is the basic modulation approach of VO_2_ film and is of vital importance for investigating the device response across the IMT without any complicated system, while the latter requires a special layout to protect the device response from distortions caused by heating circuits. As an example, Park et al. proposed a novel composite structure, as shown in Figure 4a, which consists of a combination of an asymmetric split-loop resonator (ASLR) and outer square loop (OSL) [32]. The outer square loops are designed to connect with each other to form an electrically controlled micro-heater (Figure 4b). In this way, the temperature of the VO_2_ film can be actively controlled through tuning the applied voltage on the micro-heater. The transmission spectrum of the device as a function of bias voltage is presented in Figure 4c, indicating the designed ASLR-OSL (asymmetric split-loop resonator with outer square loop) metal structure can provide effective IMT control and high-quality resonant feature simultaneously.

### 3.2. Photoinduced IMT

The optical modulation approach is of vital importance to perform nondestructive and noncontact control of the IMT and has attracted extensive research interest for its potential in all-optical communication technology. The IMT of VO_2_ film can be triggered by electromagnetic waves in the form of a continuous wave (CW) or pulsed wave over a broad wavelength range, from UV, visible and infrared to THz waves [66,68]. A study by Zhai et al. demonstrated that there are two competing mechanisms underlying the photoinduced IMT process—the slow photothermal effect and the ultrafast photodoping effect, both of which are inevitable phenomena no matter whether the incident electromagnetic wave is continuous or pulsed [100]. The mechanism of the former is still unclear and lacks systematic research, while great efforts have been made to understand the complex structural and electronic dynamics of the latter, making it a new hot issue in recent years. Generally, experiments that use the CW laser as external stimuli lack the ability to detect the ultrafast dynamics induced by the photodoping effect, and the mechanism triggering the IMT is usually explained as the photothermal effect. The measured response time in this situation varies from timescales of microseconds to seconds [33,65,66]. Except for this limitation, numerous studies have demonstrated that the CW laser can effectively modulate the IMT of VO_2_ in various THz applications and the modulation depth can be adjusted by laser intensity. The ultrafast IMT induced by the photodoping effect is mainly reported in experiments which combine the pump pulse laser with THz-TDs to provide a fs-resolution coherent investigation into the ultrafast electronic dynamics of VO_2_. Such research has demonstrated that a photoexcited IMT can be triggered by an intense fs pulse laser within 1 ps [55], promoting the emerging research on dynamically tuneable THz devices based on the ultrafast IMT of VO_2_.

#### 3.2.1. Ultrafast IMT Induced by fs Laser

As one of the simplest strongly correlated materials, the ultrafast dynamics in VO_2_ have attracted plenty of research efforts since the 2000s and have provided new insights into the physical mechanism responsible for the phase transition. Researchers have demonstrated that there a time separation exists between the IMT and SPT when VO_2_ film is triggered by intense ultrafast pump laser—the IMT occurs within 1 ps, while the SPT undergoes a much complex evolution process and takes place on a slower timescale [57]. Since the modulation phenomenon of VO_2_ film in THz range is based on the IMT, utilizing the fs laser as an excitation source enables VO_2_-based devices to respond “quasi-instantaneously.”

Ultrafast IMT of VO_2_ film could be observed via time-resolved THz spectroscopy [89]. As shown in Figure 5a, after excited by a single pulse (12 fs width) at 295 K, the THz conductivity of VO_2_ film initially increases rapidly due to the optically generated free carriers and reaches the peak amplitude at ~60 fs. Subsequently, the photoinduced carriers decay on a sub-ps timescale and the film recovers to the insulating state when the excitation fluence is lower than a critical value (Φ_c_). Only if the fs laser fluence exceeds the threshold value (Φ_c_ (295 K) = 4.6 mJ/cm^2^), long-lived photoconductivity can exist, indicating that the IMT is triggered. The resultant metallic state can last several microseconds until the heat dissipates and the film is cooled down. Figure 5b shows the THz conductivity change at 1 ps as a function of laser fluence at 295 K and 320 K. Since the directly excited photocarriers decay at this time, the conductivity change vanishes for small pump fluence but grows rapidly for pump fluence above the threshold (Φ_c_). Additionally, the fluence threshold triggering the IMT depends on the initial temperature of the sample, because heating VO_2_ film toward critical temperature helps soften the electron correlations in insulating VO_2_ film, which reduces the activation energy of the IMT [89]. As presented in Figure 5c, the fluence threshold experiences a significant reduction as the critical temperature is approached.

The time-resolved THz spectroscopy only reflects the time-resolved evolution of electronic structure, while it does not give any information directly regarding structural change. Considering that the phase transition in VO_2_ shows a high coupling of IMT and CPT, ultrafast experiments sensitive to lattice change, such as electron diffraction [52,53], X-ray diffraction [54] and coherent phonon spectroscopy [50], have been carried out to investigate the structural phase transition. Wegkamp et al. summarized their related work and gave a comprehensive picture explaining the stepwise changes throughout the phase transition process, as is shown in Figure 5d [55]. The transition could be divided into two main steps:The first step, which is several hundred femtoseconds long, is a nonthermal process. The initial photoexcited carriers change the strong electron correlation inside the V-V dimers, leading to the collapse of insulating band gap within tens of femtoseconds [51]. At the same time, the new charge distribution interacts with the lattice structure, changing the lattice potential into a non-monoclinic one [50]. The lattice potential change (LPC) represents the onset of the CPT and the subsequent atom rearrangement occurs within 300 fs, via a complex pathway, resulting in the melting of V-V dimers [53]. In conclusion, Step 1 is characterized by the formation of metastable metallic monoclinic phase.The step 2, which is around tens of picoseconds, is known as the quasi-thermal process. The excess energy of the photoinduced carriers drives the metastable monoclinic metallic structure to transform into the thermal-equilibrium rutile structure, marking the completion of the CPT. No electronic dynamics can be observed in this step, while the lattice structure continues to evolve. The resultant thermal equilibrium metallic rutile phase can maintain several microseconds due to the thermal hysteresis effect, until heat transport cools the sample down [55].

Ultrafast IMT process still works when VO_2_ film is embedded in metamaterials. Hence, tuneable THz devices based on VO_2_ film are capable of ultrafast response when excited by intense fs pulse laser.

#### 3.2.2. IMT Induced by Intense THz Field

The great improvements on the fs laser and THz generator enable short THz pulse (picoseconds) with intense field strength and pave a way to investigations on ultrafast dynamics triggered by intense THz pulse. As a strongly correlated electronic material, VO_2_ film should be able to respond to intense THz field, since the intense electric field of THz pump may disturb the electron correlation inside the V-V dimers. However, researchers have demonstrated the IMT triggered by intense THz pulse is mainly a thermal effect caused by Joule heating [67,68]. In detail, the THz electric field initially lowers the Coulomb-induced activation barrier and causes a release of carriers. Then, the newly formed carriers are accelerated by the THz electric field, leading to Joule heating via electron-lattice coupling.

Although recent advances have enabled intense THz fields with strengths as strong as 1 MV/cm, corresponding to a THz fluence of ~2 mJ/cm^2^, they are still weaker than the typical fluence threshold of photoinduced IMT [68]. Considering the requirement of stimuli strength, subwavelength resonators are integrated on VO_2_ film to locally enhance the electric field inside the resonator gaps. An example of enhanced THz field in contrast to the initial THz pump signal is shown in Figure 6a [68]. The enhancement is realized by the grid array antennas deposited on VO_2_ film with gaps of 1.5 um, as shown in Figure 6b. The geometry of the grids, designed to compromise between effective field enhancement for the THz pulse and fill a fraction of the VO_2_ film, results in, on average, four-times greater field enhancement, as shown in Figure 6c. In another related work, Thompson et al. fabricated a dynamically tuneable THz antenna by incorporating VO_2_ film with nanoslot antennas with gaps of 200 nm, as shown in Figure 6d [69]. The response of the device is shown in Figure 6e. For VO_2_ film in the insulating state, the device shows an antenna resonance at 0.9 THz, while the resonant transmission disappears as VO_2_ film is heated to the metallic state. Except for the thermal-induced modulation phenomenon, the device also exhibits a decreased transmission when the strength of the incident THz field is increased, as is shown in the time spectra in Figure 6f. Such result demonstrates the device can be modulated by enhanced THz field with the assistance of metal resonators.

In such experiments, the metal metamaterial structure is well-designed to act as both the amplifier of THz field and the plasmonic to generate resonator features, providing a viable pathway to fabricate functional nonlinear THz modulators. Although it occurs through a much different mechanism compared with the IMT triggered by the fs laser, this kind of modulation approach still reveals ultrafast response speed (picoseconds) and can be utilized for high-speed optoelectronic devices [67].

### 3.3. IMT Induced by Electric Field

The electric field is another effective approach to control the IMT of VO_2_ [60,63,119]. The underlying physical mechanism is still under debate as the roles of electric field induced doping and Joule heating are still controversial. Kalcheim et al. recently demonstrated a purely nonthermal electrically induced IMT in quasi-1D VO_2_ nanowire [120]. They successfully decoupled the nonthermal IMT process from the Joule-heating scenario. However, such phenomenon has not been reported in VO_2_ films. The more common opinion is that Joule heating may take the dominant role instead of the field-induced electron doping. To prove this point, a related work completed by Zimmers et al. is introduced here, in which an in situ measurement of film temperature across the electrically triggered IMT was performed. The local temperature inside the electrode channel was inferred according to the fluorescence spectra of the temperature-sensitive fluorescent particles, as presented in Figure 7a [61]. They proved that the resistance-temperature (*R*-*T*) curve of electrically triggered IMT overlaps with the thermally triggered one (Figure 7b), indicating that the electron doping only has negligible effects on VO_2_ film and Joule heating plays the predominant role.

To apply electric field on VO_2_ film, artificially designed electrodes are necessary [59,121]. Through changing the voltage applied on electrodes, the electric field across the channel can be controlled. The electric field threshold triggering the IMT ranges from 1.5 × 10^6^ to 2.6 × 10^6^ V/m, depending on the initial temperature and the type of the VO_2_ film [61]. However, the threshold voltage can be extremely high when using simple parallel electrodes. For instance, in early work, gold nano-slot antennas were deposited on VO_2_ film with 1 mm-wide parallel electrode, as shown in Figure 7c [122]. The large area and simple geometric design of the electrode in this device are the main reasons responsible for the high-threshold voltage (400 V) shown in Figure 7d. The fundamental way to reduce the applied voltage in electrical modulation approach is to reduce the electrode distance or minimize the fill area of the VO_2_ film [123]. For example, Zhou et al. constructed a dynamically tuneable THz device by integrating interdigitated electrodes with grid-structure VO_2_ film, as shown in Figure 7e [62]. The special geometric design in this device allows low-bias voltage control and reduces the power cost to 0.5 W. Meanwhile, the device offers a large modulation of transmitted THz waves over a broadband frequency range (Figure 7f), demonstrating that combining metallic electrodes with metamaterials is an effective way to fabricate energy-efficiency devices.

## 4. Film Deposition & Property Optimization

Researchers have demonstrated that most of the physical and chemical deposition methods, such as sputtering [74], pulsed laser deposition (PLD) [92], molecular beam epitaxy (MBE) [124], polymer assisted deposition (PAD) [125], sol-gel [126] and hydrothermal methods [127], can be utilized to synthesis VO_2_ films with high modulation performance. Moreover, the IMT properties of VO_2_ films, such as critical temperature, magnitude of THz conductivity change, excitation energy (for ultrafast IMT) and hysteresis loop width, are sensitive to the oxidation states and microstructures of VO_2_ film. As a result, it is possible to modify these properties in the synthesis process for different applications. The recent efforts in this field have focused on how to reduce the energy consumption used to trigger the IMT without sacrificing modulation performance. One of the effective methods is ion doping. For instance, researchers have reported doping W^6+^ ion into VO_2_ film could not only lower the critical temperature toward RT [91] but could also reduce the pump fluence threshold for ultrafast IMT [128]. However, a certain degree of degeneration in modulation performance can be observed accompanied by W^6+^ doping. Another approach involves introducing epitaxial strain to influence the microstructures of VO_2_ film. This approach highly relies on epitaxial growth techniques and avoids the degradation of modulation phenomenon [129]. Other optimizations, such as the broadening of the phase transition temperature window [91] and anisotropic modulation [130], can also be realized by controlling the synthesis process of VO_2_ film. The modification of IMT properties via deposition techniques provides more freedom for practical applications, extending the applicability of VO_2_ film in tuneable THz devices.

### 4.1. Ion Doping

Reducing the energy consumption for triggering the IMT is of critical importance for practical applications. For example, reducing the critical temperature, laser fluence threshold or electric field threshold of IMT helps reduce the energy cost of VO_2_-based THz devices and benefits the simplification of excitation unit. Researchers have demonstrated that transition metal ions, including but not limited to Nb^5+^, Mo^6+^, W^6+^ and hydrogen ion H^+^, can effectively reduce the critical temperature of VO_2_ [46,128,131,132,133]. Among them, W^6+^ is the most effective and commonly used. Figure 8a presents a typical X-ray photoelectron spectra (XPS) of W-doped VO_2_ film, in which the existence of W element is confirmed by the characteristic 4f peaks of W^6+^ ions [101]. Thermal studies on THz transmission change (Figure 8b) have demonstrated that doping W ions into VO_2_ film can reduce the critical temperature of IMT with a rate of ~22 ± 4 °C/at.%W and broaden the phase transition temperature window [91]. As for the ultrafast IMT induced by the fs laser, Émond et al. reported that the fluence threshold of W_0_._013_V_0_._987_O_2_ film is reduced to 1.1 mJ/cm^2^, down from the 3.8 mJ/cm^2^ in pure VO_2_ film, as shown in Figure 8c,d [128]. These experiments demonstrate that W doping help reduce the requirement for triggering the IMT in both thermal and optical approaches.

### 4.2. Epitaxial Growth Techniques

Except for ion doping, another effective approach to optimize the IMT properties is to alter the film structure through substrate influence. Recent research reported by Liang et al. took a novel approach to reduce the excitation energy of ultrafast IMT [129]. They deposited Van der Waals (vdW) heteroepitaxial VO_2_ film on ultrathin (~13 um) mica substrate. The schematic of the film-substrate interface is in shown in Figure 8e. The pump fluence threshold (0.21 mJ/cm^2^) of the vdW-epitaxial VO_2_ film, extrapolated from the THz transmittance curve shown in Figure 8f, is only ~5% of the traditional epitaxial film. Meanwhile, the vdW epitaxial film exhibits excellent modulation effect—the transmittance change reaches 81.2% as the IMT is triggered. They attributed the significant reduction in fluence threshold to the impact of vdW heteroepitaxy. Typically, the bonding strength of vdW heteroepitaxy is 0.1–10 kJ/mol, much lower than the strength of chemical bonding (100–1000 kJ/mol). A schematic illustration of the difference between chemical bounding and vdW bounding is presented in Figure 8g. Since the IMT of VO_2_ is accompanied by a large modification in the lattice structure, the strong chemical bonding on the traditional epitaxial interface will give rise to an intense clamping effect and thus cause a barrier for phase transition. In contrast, weak film–substrate interaction on the vdW epitaxial interface significantly reduces the influence caused by substrate clamping effect, resulting in a reduction in excitation energy for the IMT. Additionally, the poor heat conduction in the vdW epitaxial interface prevents heat from transferring to substrate, improving the energy efficiency of the pump laser. Both factors are considered responsible for the significant fluence threshold reduction [129,134]. 

IMT properties of the traditional epitaxial VO_2_ films are sensitive to the interfacial strain induced by substrate mismatch. Researchers have demonstrated that the THz properties of epitaxial VO_2_ films are different when deposited on different substrates [135]. For example, VO_2_ films deposited on m- and r-sapphire substrates reveal relatively lower critical temperature and higher modulation depth compared with the films on c-sapphire substrate [74,90,134,135,136].

Generally, VO_2_ film exhibits isotropic THz conductivity in directions parallel and perpendicular to V-V chains (c_R_ axis). However, the symmetry can be altered by synthesizing epitaxial VO_2_ film on a-cut TiO_2_ substrate. The surface morphology of the sample is presented in Figure 9a, in which periodic buckling and cracking paralleling to the *c*_R_ axis can be observed. The lattice mismatch between VO_2_ film and TiO_2_ substrate results in tensile strain along the *c*_R_ axis, and compressive strain along the *a*_R_ axis and *b*_R_ axis, causing the highly oriented THz transmission properties shown in Figure 9b,c [130]. Remarkable difference could be observed in the transmission spectrum shown in Figure 9b—the THz transmission decreases by ~85% along the *c*_R_ axis and by ~15% along the *b*_R_ axis after the IMT. Temperature-dependent THz conductivity in the heating and cooling cycles also exhibits significant anisotropic features (Figure 9c). The uniaxial modulation phenomenon along the *c*_R_ axis, accompanied by the large modulation depth, can be the basis for orientation-related applications in the THz regime.

## 5. Dynamically Tuneable THz Devices Based on VO_2_

VO_2_ film is a natural THz amplitude modulator in itself, since the IMT of VO_2_ results in a THz conductivity change of several orders of magnitude. More importantly, the modulation depth of VO_2_ film is highly tuneable due to the phase coexistence phenomenon during the IMT. Multistate THz response can be realized in VO_2_ film through tuning the strength of external stimuli, promoting its applications in fields such as antireflection coating [90], impedance matching [137,138] and multistate optical memorizers [33,35].

Additionally, the advances in micromachining technologies make it possible to integrate high-quality VO_2_ films into metamaterials. In this way, VO_2_ can be coupled with functionalized metamaterials to fabricate tuneable THz devices. Metamaterial is a kind of artificially designed material consisting of sub-wavelength plasmonic micro/nanostructures and has been demonstrated as an effective tool to manipulate the electromagnetic properties of THz waves, such as propagation direction, amplitude, phase and polarization. Since the resonance feature of metamaterial is sensitive to the surrounding dielectric environment, integrating phase-change material into resonators enables the metamaterial to be dynamically controlled by external stimuli.

Furthermore, unique optical memory-type function can be achieved based on the intrinsic hysteresis behaviour of the IMT. By utilizing external thermal, optical or electrical stimuli, stationary metallic state can be written into VO_2_ film and then read through THz transmission response, which is the basis for rewritable memory devices.

### 5.1. VO_2_ Hybrid THz Metamaterial

On the one hand, despite the fact that fabrication techniques can introduce some novel features, the functionality of pure VO_2_ film is still limited by its intrinsic physical properties. On the other hand, metamaterials can effectively manipulate the state of propagating THz waves but are unable to be dynamically controlled without any active designs or materials. In this case, incorporating metamaterials with VO_2_ presents a potential for functionalized and controllable THz modulators. To take full advantage of the phase-change phenomenon of VO_2_, researchers have replaced or filled the key component of metamaterials with continuous VO_2_ film or VO_2_ pieces. Once the optical constant of VO_2_ is affected by external stimuli, the dielectric environment of the metamaterial will be changed, and then the response even functionality of the metamaterial will be modified.

In this section, dynamically tuneable metamaterials based on VO_2_ film are introduced. To better understand the functionality enabled by VO_2_, a classification of the device structure of VO_2_ hybrid metamaterials is provided, which divides them into VO_2_ metamaterial, metal-metamaterial/VO_2_ film and metal VO_2_ hybrid metamaterial. The applications of different structures are introduced, along with the advantages and limitations.

#### 5.1.1. Metamaterials Made of Pure VO_2_

One of the simplest designs of VO_2_-based tuneable metamaterials is to directly utilize VO_2_ as resonators. When the VO_2_ meta-atoms are in the insulating state, the device is transparent to the incident THz waves. Only if the IMT is triggered, the VO_2_-fabricated resonators begin to operate. In this way, “on-off” switching between transparent state and resonator operating state can be realized [66,98,139]. However, although VO_2_ film undergoes a remarkable transition in THz conductivity by several orders of magnitude, the film remains somewhat transparent to THz waves even in its metallic state. To ensure enough modulation depth, THz metamaterials made of pure VO_2_ requires larger film thickness (~1 um) compared with those made of metal (~200 nm) [140]. It should be noticed that VO_2_ film with micron-level thickness is hard to fabricate, limiting the applications of THz metamaterials made by pure VO_2_. For example, Wen et al. fabricated an active THz metamaterial by directly using patterned polycrystalline VO_2_ film as cut-wire resonators (Figure 10a) [139]. The temperature-dependent frequency spectrum of the device is shown in Figure 10b, demonstrating a switching between the high-transparent state and the resonant sate. Across the IMT, the modulation depth of transmission amplitude reaches 65% at the resonant frequency (0.6 THz). The thickness of the VO_2_ film utilized in this work was 800 nm, far exceeding the average thickness (150–300 nm) reported in other metal VO_2_ hybrid metamaterials. In another example, super-thick VO_2_ film (1.2 um) was utilized as coating layer of silicon columns to fabricate state-converter-plasmonics (SCP), as shown in Figure 10c [66]. The SCP can be controlled by CW laser and the modulation depth up to 70% is achieved over a broad frequency range (Figure 10d).

#### 5.1.2. Metal Metamaterial Deposited on VO_2_ Film

Utilizing continuous VO_2_ film as the substrate layer of metal metamaterial is a straightforward method to fabricate tuneable THz optics [140,141]. Such devices can be controlled to switch between two discrete states. When the VO_2_ film is in the insulating state, the VO_2_ film layer is relatively transparent to THz waves and the device response is determined by the embedded metal resonators. After the phase transition is triggered, THz waves will be reflected by metallic VO_2_ film and the resonators no longer operate. As an example, Shin et al. fabricated a tuneable linear polarizer by depositing metal gratings on VO_2_ thin film. The structure of the device is shown in Figure 11a [142]. The temperature=dependent frequency spectrum (Figure 11b) demonstrates an improved modulation phenomenon, since the metal gratings greatly enhance the cut-off effect by nearly an order of magnitude when the VO_2_ film is in the metallic state. Meanwhile, the original linear polarization character of metal gratings with a polarization degree up to ~0.985 can be observed in this composite device, as shown in Figure 11c, which is sufficient for use as a linear polarizer. Another similar application is the tuenable meta-surface lens which consists of a tri-layer structure, including gold V-shaped antennas, a VO_2_ thin film layer and sapphire substrate, as presented in Figure 11d,e [143].

Dynamically tuneable focal intensity can be realized through tuning the temperature of VO_2_ film. The evolution of the amplitude distribution in the focal plane in the heating process is shown in Figure 11f. The focal spot initially holds the strongest energy at 293 K when VO_2_ film is in the insulating state. As the critical temperature is approached, the focal intensity gradually weakens and is finally reduced to zero after the IMT is completed.

Since the transmittance change of VO_2_ film is limited by its intrinsic properties, realizing higher modulation depth with controllable frequency range is an important issue. As an example, Choi et al. fabricated a band-pass THz modulator by depositing gold nano-slot antenna pattern on the top of VO_2_ film (Figure 12a) [24]. When the VO_2_ film is in the insulating state, the device reveals almost perfect transmission at 0.5 THz due to the strong funnelling effect of nano-resonator (Figure 12b). Once VO_2_ film transforms to metallic state, nano-resonators will be electrically shorted and THz transmission will switch to cut-off state (Figure 12d). The extinction ratio at 0.5 THz, defined by the transmission maximum to minimum signal strength, improves from 10 in bare VO_2_ film to 10^5^ in patterned VO_2_ film, as shown in Figure 12c. However, the bandwidth of this device is limited by the sharp resonant features. In order to realize high extinction ratio and broadband modulation in a single device, a multi-antenna structure constructed by a series of antenna-slots with different geometric dimensions is deposited on VO_2_ film, as shown in Figure 12e [23]. The corresponding transmission spectrum is shown in Figure 12f. Complete switching with extinction ratio up to 10^4^ over an ultra-broadband frequency range can be realized.

#### 5.1.3. Metal VO_2_ Hybrid Metamaterial

Since the high-reflection of metallic continuous VO_2_ film limits the functionality of VO_2_-based metamaterial, replacing VO_2_ film with VO_2_ pieces presents much more freedom in device response. The novel functions of metamaterials, such as frequency selection and polarization conversion, originate from the resonant features of the sub-wavelength structure and are sensitive to the changes in material property. In this way, phase-change material with a small fill fraction can give rise to large modification in device response. Additionally, reducing the phase-change area of the tuneable devices helps decrease the energy consumption and is of vital importance for practical applications in requirement of low-bias control.

For example, by embedding VO_2_ pieces as link bridge in loop cross dipole (LCP), Zhu et al. fabricated a band-pass filter with tuneable centre frequency. The schematic and optical microscope images of the VO_2_ hybrid LCP are shown in Figure 13a,b, respectively [30]. When VO_2_ components are heated to the metallic state, the effective length of the LCP changes. As a result, the frequency centre of the resonant peak shifts from ~0.41 THz to ~0.54 THz, as is shown in the transmission spectrum (Figure 13c). Another example is a tuneable phase shifter controlled by CW laser (Figure 13d) [26]. The device consists of a composite photoconductive structure (PCS), a combination of dipole resonance (short wire), VO_2_ metal hybrid capacitive inductance resonance (split ring) and long metallic wire (Figure 13e). The L-C resonance and dipole resonance are coupled together to enhance the phase jump triggered by the IMT of VO_2_ gap. As a result, a phase shift up to 130° within 55 GHz bandwidth can be realized in this phase converter, as shown in Figure 13f.

Integrating VO_2_ with metamaterial can even induce switching between different functionalities. For example, Ding et al. suggested a multifunctional device with the ability to switch between a broadband absorber and a reflecting half-wave plate [25]. The schematic of the device is presented in Figure 13g, which is characterized by a multilayer structure. From top to bottom, the multiple-layer device consists of a rectangular VO_2_ antenna array, chromium dual square resonators, continuous VO_2_ film and a chromium substrate. When VO_2_ is in the insulating state, the VO_2_ antenna array and continuous film are highly transparent to incident THz waves and the reflection spectrum is determined by the square resonators and chromium substrate, resulting in a broadband absorber state (Figure 13h). After the IMT is triggered, the VO_2_ components begin to work and the device is switched into a polarization converter. The corresponding simulated reflection spectrum is shown in Figure 13i, indicating the incident linear polarized THz waves are converted into cross-polarized reflected waves with a conversion rate up to 60% in the range from 0.6 THz to 1.2 THz.

In conclusion, VO_2_ film can be successfully integrated into diverse THz metamaterials to provide dynamic modulation capability for a variety of applications. The dynamic performance enabled by VO_2_ is closely related to the fill fraction of the phase-change area. For continues VO_2_ film, when the IMT is triggered globally, the whole device will be transformed into a high-reflection state [140,141,142,143]. However, in some cases, the IMT of continuous VO_2_ film can be locally triggered by THz pulse or electric field with the assistance of integrated electrodes or resonators [59,67,68,69]. The locally triggered phase transition, as well as the straightforward discretely distributed VO_2_ pieces [35], can provide much more freedom for device response. It not only supports continuous tuning of electromagnetic properties such as the polarization degree [28], centre frequency [30] and phase shift [26], but also allows switching between different functionality, for example, switching between a broadband absorber and a reflecting half-wave plate [25].

### 5.2. Optical Memory

For VO_2_, the so-called “memory effect” signifies the persistence of metallic state after external stimuli are turned off [123]. Such a phenomenon, accompanied with the remarkable difference in material properties between insulating and metallic states, can be the basis for rewritable memory-type applications. Additionally, because the effective THz conductivity of VO_2_ film depends on the phase fraction of metallic domains and the metallic phase fraction depends on the strength of external stimuli, distinguishing multiple states can be recorded in VO_2_ film by varying the strength of external stimuli. The recorded information can be read through the response of THz waves and erased by cooling the phase-change area down. All these memory operations, including writing, reading and erasing, can be performed by all-optical approaches.

As an example, memory operations of an all-optically driven 2-bit memorizer made by simple VO_2_ film have been investigated [33]. In this all-optical memory system, an intense fs laser pulse is used as writing channel while a CW laser provides a continuous bias power and is constantly switched on except for the erase operation (Figure 14a). Ultrafast IMT enabled by the intense fs laser results in a “quasi-simultaneous” transition in THz transmittance, allowing the recorded state to be read out as soon as the information is written in. The ground “00” state of VO_2_ film maintained by the CW laser (P_0_) is the start stage of IMT, so that the THz transmission can respond to a single fs pulse (100 fs, 390 μJ). The four discrete states, denoted as “00,” “01,” “10” and “11” shown in Figure 14b, correspond to the record of zero, one, two and three fs pulses, respectively. The “erase” operation is performed by turning the CW laser off for 2 s and it takes ~3 s for VO_2_ film to recover to the thermal-equilibrium ground state.

In another example (Figure 14c), through filling the gaps of asymmetric split-ring resonators (ASRRs) with VO_2_ pieces, an electrically controlled low-bias THz memorizer was fabricated [35]. The metal structure of the device, a combination of ASRR array and long metal lines, plays a dual role by also providing a turn-on current and manipulating the frequency response of the propagating THz waves. The frequency response of the device as a function of the applied current is shown in Figure 14d, demonstrating that the THz transmission of the metadevice is highly tuneable. The timing diagram of the binary coding process is shown in Figure 14e. The ground state “0” is maintained by a continuous current (0.58 A), and the “write” (1 A, 1 s) and “erase” (0 A, 2 s) pulses are implemented to switch the device between the “0” and “1” states. As a result, unambiguous memory effect can be observed in the THz transmission diagram at 0.63 THz. The authors also investigated the multistate memory operation of the metadevice, as shown in Figure 14f. Through tuning the strength of “write” current pulse, four distinguishing states coded as “00”, “01”, “10” and “11” could be written into the metadevice.

Compared with the aforementioned memorizer made by pure VO_2_ film, the coupling of metamaterial with VO_2_ pieces results in great improvement in practicability. It not only enhances the contrast between different states but also makes a significant simplification of the memory operations, presenting great potential for memory-related applications in the THz regime. 

## 6. Summary and Outlook

All of the VO_2_-based dynamically tuneable THz devices encompass three fundamental elements: The intrinsic properties of VO_2_ film, external stimuli for active control of IMT and device structure which decides the functionality and operating frequency.

The first element, the intrinsic properties of VO_2_ film, greatly affects the modulation depth and energy consumption of the device. Generally, VO_2_ film exhibits reversible IMT behaviour in response to external stimuli, yielding remarkable changes in THz conductivity. This modulation phenomenon is closely related to the chemical and crystalline structure of VO_2_ film and can be optimized in the deposition process, with, for example, ion doping and epitaxial growth. One of the related hot issues is to reduce the energy consumption of the IMT, namely, lowering the phase-change temperature to RT or reducing the stimuli threshold triggering the IMT. If room-temperature and low-energy-consumption control of the IMT is possible, it may lead to significant enhancement of device stability and response speed. In view of this, the vdW-epitaxial VO_2_ film proposed by Liang et al. is of vital importance, since it reduces the laser fluence threshold of IMT to 0.21 mJ/cm^2^ (~2.1 mW) at RT, a value only ~5% of the normal films, and is sufficiently low for practical application. Additionally, the ultra-thin mica substrate (13 um) applied in this study avoids disruptions from the Fabry–Pérot effect and helps to reduce the insertion loss of the whole film device, both of which are fascinating features for THz optics.

The second element, external stimuli, determines the responding speed of the device. VO_2_ can respond to various stimuli, but only a few of them can be coupled with THz devices, including thermal, optical (CW laser, fs laser and intense THz field) and electrical excitations or any combination of them. The very different excitation approaches act initially on the VO_2_ in very different ways, but most of them, except for the fs pump laser, eventually produce a thermal effect that gradually accumulates to push the IMT thermally. Since it takes time for thermal accumulation, the responding time of such approaches varies over a large range of timescales, from picoseconds to several minutes, depending on the strength and duration of excitation, as well as the initial temperature and thermal mass of the VO_2_ component. In the case of nonthermal IMT triggered by fs laser, ultrafast photo-response originating from direct photoexcitation effect has been demonstrated, and the responding time evolves to hundreds of femtoseconds to several picoseconds, which makes VO_2_ a promising material to fabricate high-speed THz modulators. Figure 15 illustrates an overview of the VO_2_-based THz devices regarding the responding time, clearly showing the timescale and underlying mechanism of different modulation approaches.

The last element, the structural design, determines the functionality and operating frequency of the device. The simplest structure, composed of a pure VO_2_ film, can act as a broadband amplitude modulator, with a modulation depth up to 85%. Other designs include planar metamaterial fabricated by pure VO_2_, metal-metamaterial deposited on VO_2_ film and VO_2_ metal hybrid metamaterial. Novel electromagnetic features, such as frequency selection, phase shifting and polarization converters, can be realized based on structural design, and such features can be dynamically tuned by applying external stimuli on the VO_2_ component. Additionally, optical memory operation based on the intrinsic hysteresis behaviour of IMT has been demonstrated both in simple VO_2_ film and metal-VO_2_ hybrid metamaterials.

According to the aforementioned three key elements, we favour three directions of VO_2_-based active THz device development: (i) Tuning the IMT properties of VO_2_, such as reducing the critical temperature, decreasing the excitation energy for ultrafast IMT and improving modulation depth; (ii) improving the modulation approach to obtain high-speed and high-precision control of IMT; (iii) enhancing functionality, such as developing intelligent metamaterials with programmable electromagnetic response.

Nowadays, the research of metamaterials is no longer limited to a fixed, static electromagnetic response. New issues are about tuneable, reconfigurable and programmable metadevices with greater functionality and applicability [37,38,144,145,146]. VO_2_ has shown a great potential in this new field. Since the IMT of VO_2_ can be locally triggered by optical and electrical methods [59], metamaterials with VO_2_ as key component allow programmatic control of each unit cell, which is the basis for intelligent THz devices. Such applications have already been demonstrated in infrared frequency range [147,148] but are still absent in the THz regime. Meeting this challenge can extend the application scope of tuneable metamaterial based on VO, and may give rise to next-generation THz devices.

## Figures and Tables

**Figure 1 nanomaterials-11-00114-f001:**
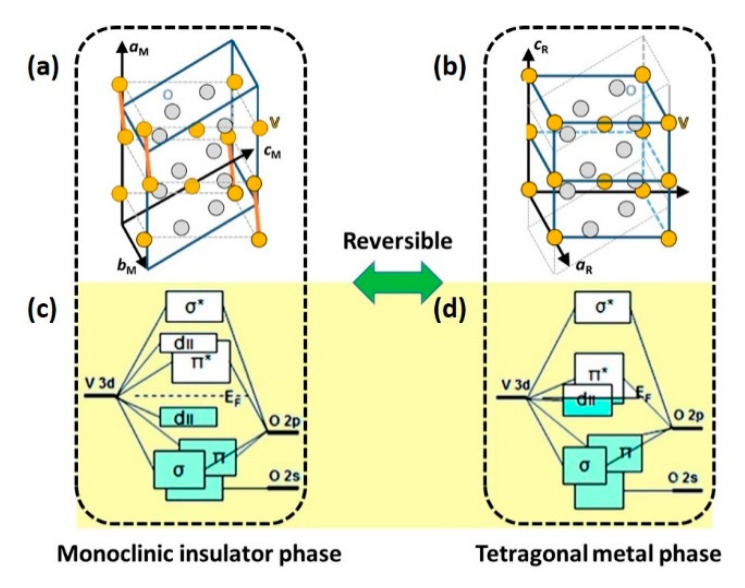
Schematic of the crystal structure of VO_2_ in the (**a**) insulator (monoclinic) and (**b**) metallic (tetragonal) phase. V atoms: Orange balls; O atoms: Grey balls. Schematic of the band scheme of (**c**) VO_2_ (M) and (**d**) VO_2_ (R) based on crystal field model. Reproduced from [88], with permission from American Chemical Society, 2011.

**Figure 2 nanomaterials-11-00114-f002:**
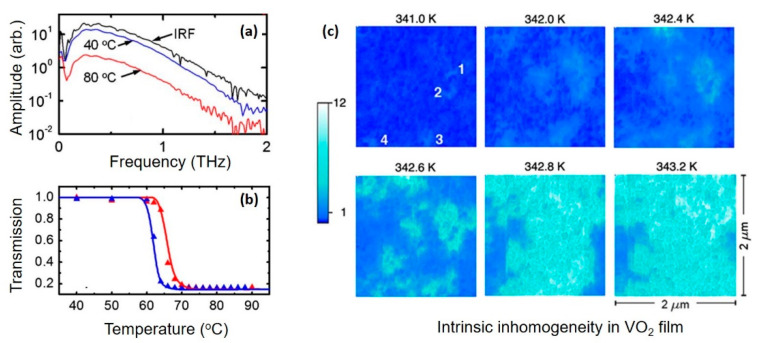
Modulation phenomenon in the terahertz (THz) regime. (**a**) THz transmission spectrum of VO_2_/*r*-sapphire sample at low (40 °C) and high (80 °C) temperatures. THz transmission through air is also shown to illustrate the instrument response function. (**b**) Normalized (to T = 40 °C) THz field amplitude transmission as a function of the temperature (symbols) for VO_2_ films grown on c-sapphire. Reproduced from [90], with permission from Optical Society of America, 2012. (**c**) The near-field scattering infrared microscopy pictures over the same 2 μm by 2 μm area (infrared frequency ω = 930 cm^−1^) in heating process. The spatial resolution is 15 nm. The metallic regions (light blue) give higher scattering near-field amplitude compared with the insulating phase (dark blue). Four newly formed metallic puddles are marked as 1, 2, 3 and 4 on the T = 341.0 K image. Reproduced from [93], with permission from American Physical Society, 2009.

**Figure 3 nanomaterials-11-00114-f003:**
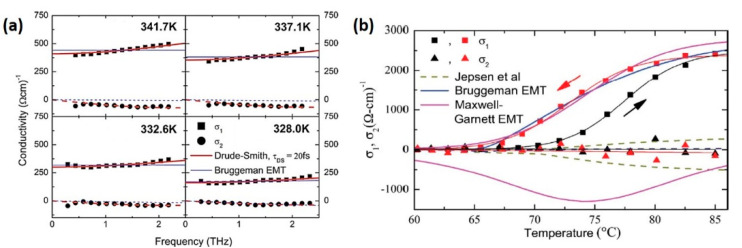
Predictions of Drude-Smith model and effective medium theory (EMT) model. (**a**) Complex conductivity of VO_2_ thin film on a-sapphire substrate at different temperatures as the sample is cooled. Fits are for the Drude-Smith model (thick red lines) and Bruggeman EMT (thin blue lines). Reproduced from [77], with permission from American Institute of Physics 2010. (**b**) Complex conductivity of the VO_2_ thin film grown on c-sapphire as a function of temperature during heating (black symbols) and cooling (red symbols). The blue and magenta curves show the predictions of the Bruggeman EMT and MG EMT, respectively. Reproduced from [73], with permission from Optical Society of America 2011.

**Figure 4 nanomaterials-11-00114-f004:**
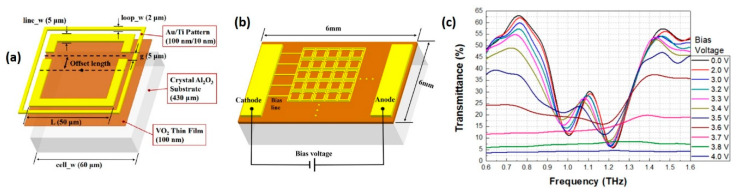
Insulator-metal transition (IMT) controlled by integrated micro-heater. (**a**) Schematic of the asymmetric split-loop resonator with outer square loop (ASLR-OSL) based on VO_2_ and (**b**) configuration of a bias engagement for the electrical active-control of the ASLR-OSL. (**c**) Frequency responses with increasing bias voltage. Polarization of THz waves is parallel to the gap. Reproduced from [32], with permission from Optical Society of America 2018.

**Figure 5 nanomaterials-11-00114-f005:**
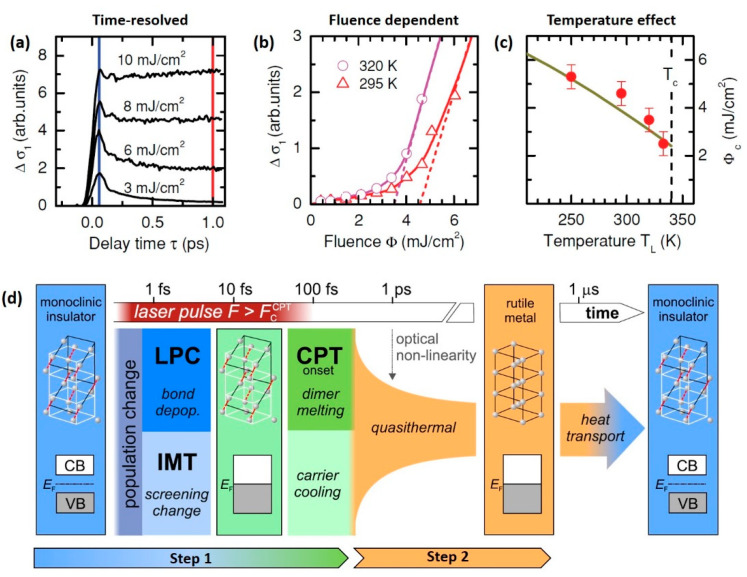
Ultrafast IMT triggered by pulse pump laser. (**a**) THz conductivity change (Δσ_1_) of polycrystalline VO_2_ film (120 nm-thick, deposited on diamond substrate) after excited by a 12 fs, 800 nm laser pulse at 295 K for different pump fluence. (**b**) Extrapolation of Δσ_1_ (at 1 ps) curves (red triangles: 295 K, magenta circles: 320 K) to a critical fluence of Φ_c_ (295 K = 4.6 cm^2^ and Φ_c_ (320 K) = 3.5 mJ/cm^2^, respectively). (**c**) Dependence of threshold fluence Φ_c_ on lattice temperature T_L_. Reproduced from [89], with permission from American Physical Society 2011. (**d**) Comprehensive picture of the ultrafast dynamics in VO_2_ film during the photoinduced phase transition. Optical excitation uses ~60 fs, 800 nm pump laser pulse above $F_{C}^{CPT}$ (fluence threshold for crystallographic phase transition). The main two steps: (1) Characterized by metastable metallic state with monoclinic lattice structure. Initially, the screening of the coulomb interaction (CIA) is disturbed, leading to the IMT on screening timescales (few fs). On the same timescale, V–V bonding orbitals are depopulated by photoexcitation [51], leading to the lattice potential change (LPC) [50]. The resulting transient phase of vanadium dioxide in this stage is a highly excited metal with monoclinic atom arrangement. The crystallographic phase transition (CPT) subsequently happens characterized by the melting of the V-dimers [53], concurrently with the relaxation of excited electrons and holes (carrier cooling) in the metallic band structure [51]. (2) Then, the system evolves quasi-thermally to its high-temperature equilibrium rutile metallic phase in ~300 ps, marking the completion of IMT. The photoinduced rutile metal phase maintains until sufficient heat has been transported away, on a timescale varying from several microseconds to hundreds of microseconds depending on the local thermal diffusivity. Reproduced from [55], with permission from Elsevier 2015.

**Figure 6 nanomaterials-11-00114-f006:**
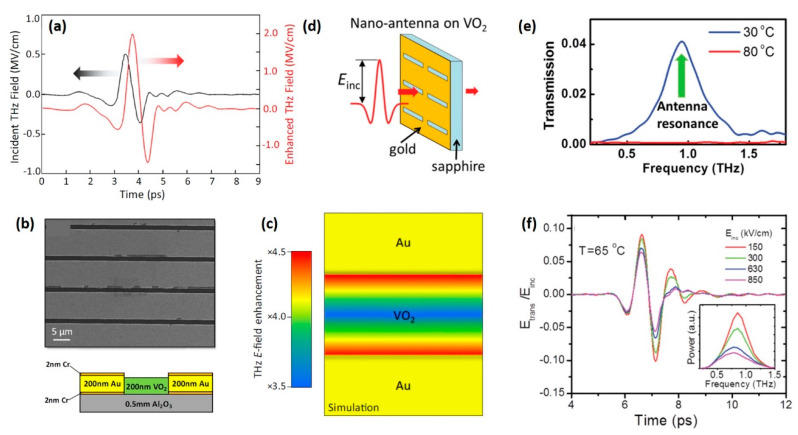
IMT triggered by intense Terahertz pulse. (**a**) Simulated enhanced THz field (red) in the metamaterial gaps using experimental data (black) as the input. (**b**) SEM image of the grating pattern showing wider Au lines (8.5 μm) and narrower (1.5 μm) gaps exposing the VO_2_ film (dark contrast). The cross-sectional profile of the structure is shown below. (**c**) THz field enhancement as a function of position within the 1.5 μm gap, showing an average field enhancement of ×4. Reproduced from [68], with permission from American Physical Society 2018. (**d**) Gold nanoantenna array (antenna width, 200 nm; length, 60 μm) is patterned on a 100 nm-thick VO_2_ thin film deposited on sapphire substrate. (**e**) A transmission spectrum showing antenna resonance at 0.9 THz with insulating VO_2_ at room temperature (blue line), whereas the resonant transmission disappears with metallic VO_2_ at high temperature (red line). (**f**) Response of the device under different THz field strengths of 150 kV/cm, 300 kV/cm, 630 kV/cm and 850 kV/cm at 65 °C. The corresponding frequency spectra are shown in the insets. Reproduced from [69], with permission from American Chemical Society 2015.

**Figure 7 nanomaterials-11-00114-f007:**
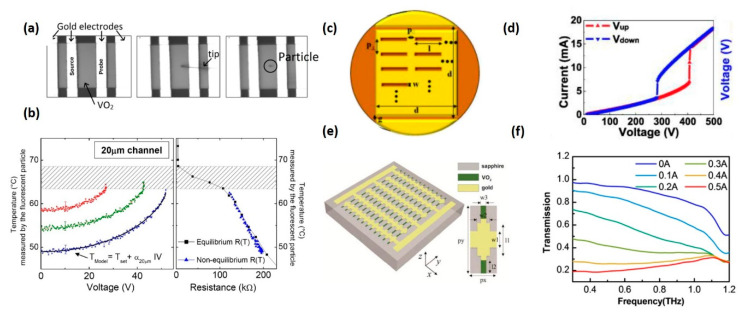
Electrical field-excited IMT in VO_2_ thin film. (**a**) 20 um channel before, during, and after positioning the micron-wide rare-earth fluorescent particle sensor. The temperature of the particle is measured through fluorescence spectroscopy. (**b**) Local temperature versus DC voltage (**left**) and local temperature versus resistance (**right**) in voltage-induced IMT (blue triangle) and thermal equilibrium IMT (black square). Reproduced from [61], with permission from American Physical Society 2013. (**c**) Schematic of the nanoantenna array deposited on VO_2_ film with 1 mm-wide parallel electrodes. (**d**) External voltage-driven insulator–metal transition in VO_2_ thin film. The voltage is increased at a rate of 1 V/s. Reproduced from [122], with permission from Optical Society of America 2011. (**e**) The schematic structure and a unit cell of the low bias controlled VO_2_ hybrid metasurface. The device consists of metal bias lines arranged with grid-structure-patterned VO_2_ film on sapphire substrate. (**f**) Magnitude transmission with different electrical biases under a constant heating temperature of 68 °C. Reproduced from [62], with permission from Optical Society of America 2017.

**Figure 8 nanomaterials-11-00114-f008:**
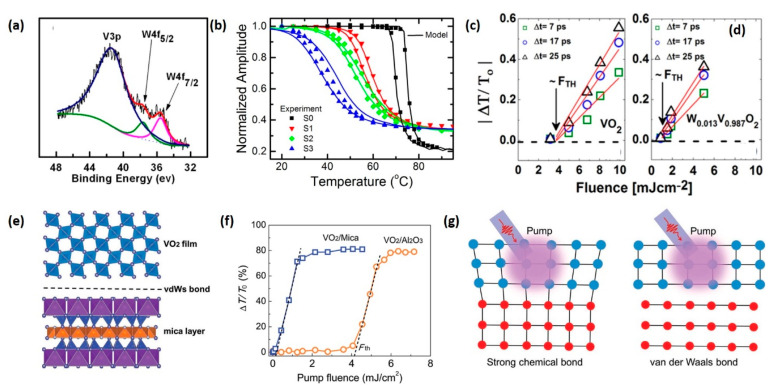
The influence of fabrication techniques. (**a**) XPS pattern of W-doped VO_2_ film. Reproduced from [101], with permission from American Institute of Physics 2015. (**b**) Normalized THz transmission (symbols) as a function of temperature for W_x_V_1-x_O_2_/sapphire samples S0, S1, S2 and S3 (x = 0, 1.47%, 1.59% and 1.73% respectively). Reproduced from [91], with permission from American Institute of Physics 2014. (**c**,**d**) Dependence of THz transient transmission variation on excitation fluence at different delay times (Δt) of 7 ps, 17 ps and 25 ps for (**c**) VO_2_ and (**d**) W_0_._013_V_0_._987_O_2_ films, respectively. Reproduced from [128], with permission from American Institute of Physics 2017. (**e**) Schematics of Van der Waals (vdW) heteroepitaxial VO_2_ film deposited on ultrathin mica substrate. (**f**) Pump fluence dependence of differential transmission (ΔT/T_0_) signals of VO_2_/mica and VO_2_/m-sapphire films at 1 THz. (**g**) Schematics of laser-induced lattice changes of heteroepitaxial VO_2_ films on substrates with strong chemical bonding (covalent or ionic) and vdW bonding. Reproduced from [129], with permission from WILEY-VCH.

**Figure 9 nanomaterials-11-00114-f009:**
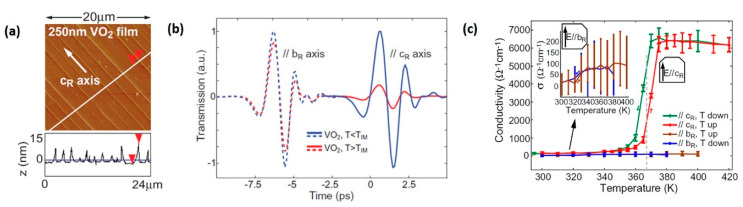
Anisotropic IMT through epitaxial strain engineering. (**a**) AFM phase image (0°–5° scale) and corresponding height profile of the 250 nm VO_2_/TiO_2_ sample shows buckling-induced ridges along the *c*_R_ axis. (**b**) THz transmitting waveforms along the *c*_R_ and *b*_R_ axis, below (blue) and above (red) the critical temperature T_MI_, normalized to the room-temperature value. The modulation depth along the *b*_R_ axis (~15%) is dramatically different from that along *c*_R_ (~85%). (**c**) Temperature dependence conductivity along the *c*_R_ axis and *b*_R_ axis in heating and cooling cycles. Reproduced from [130], with permission from Institute of Physics 2012.

**Figure 10 nanomaterials-11-00114-f010:**
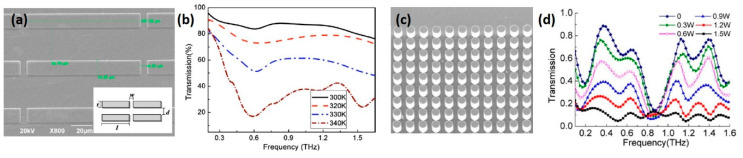
Tuneable metamaterials made of patterned VO_2_ thick film. (**a**) SEM image of the VO_2_ cut-wire array with measured dimensions of l = 107.25, w = 6.25 and t = 13.25 (all units in um). (**b**) Temperature-dependent THz transmission curves for cut-wire metamaterial. Reproduced from [139], with copyright from American Institute of Physics 2010. (**c**) SEM image of the state-conversion-plasmonics (SCP) consisting of silicon columns with VO_2_ coating. (**d**) Transmission spectra of the SCP with different pump powers (532 nm CW laser) under the double 45° tilted pumping. Reproduced from [66], with permission from Optical Society of American 2013.

**Figure 11 nanomaterials-11-00114-f011:**
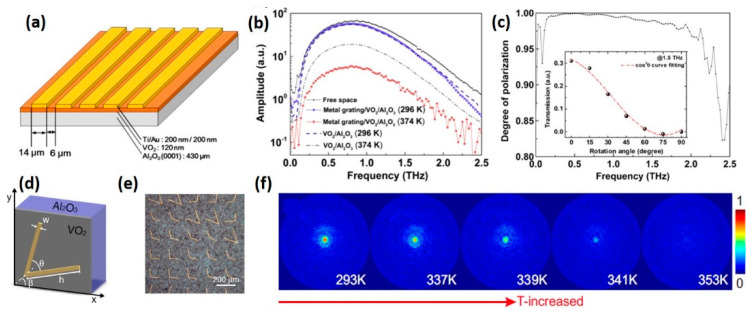
Tuneable metamaterials consisting of VO_2_ film and metal meta-atoms. (**a**) Schematic of the switchable linear polarizer: Au/Ti metallic gratings deposited on VO_2_/c-sapphire. Metal gating width: Free space = 14 μm:6 μm. (**b**) Frequency responses of the metal-grating hybrid film and bare film in 0–2.5 THz range at 296 K and 374 K. (**c**) The degree of polarization. Inset: Polarizer angle rotation dependence of wave intensity at 1.5 THz (dot line) and cos^2^(θ) term fitting curve (red dash-dot line). Reproduced from [142], with permission from Institute of Physics 2015. (**d**) Schematic of the tuneable metamaterial lens: View of a single V-shaped antenna unit. (**e**) SEM image of the partial metamaterial. (**f**) The evolution of the amplitude distribution on the focal plane in the heating process. Reproduced from [143], with permission from Springer Nature 2016.

**Figure 12 nanomaterials-11-00114-f012:**
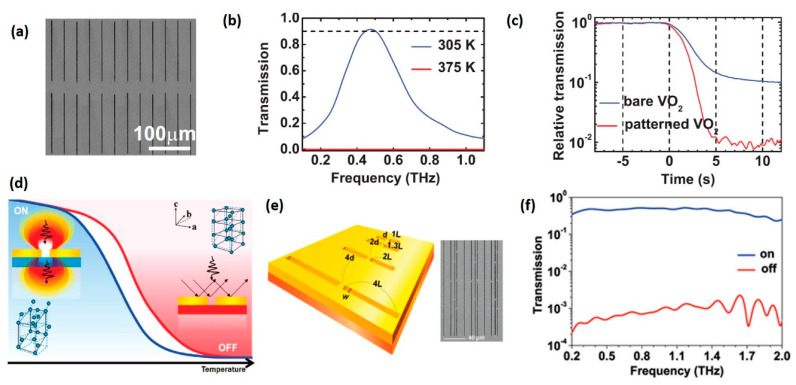
THz modulator with high extinction ratio. (**a**) SEM image of the nanoslot-antenna/VO_2_ film sample. (100 um period, 150 μm length and 450 nm width). (**b**) Transmission spectra at the insulator and metal state of VO_2_. (**c**) THz switching time measurements for the bare (blue line) and patterned (red line) samples when the structural phase transition of VO_2_ is driven by thermal heating. Reproduced from [24], with permission from American Institute of Physics 2011. (**d**) Phase transition diagram of the nanopatterned VO_2_ thin film as a function of temperature, based on the THz transmission. (**e**) Schematic of broad-band gold resonator patterns on a VO_2_ thin film (left). SEM image (right) of a nanoresonator pattern sample (350 nm width and 50 μm, 65 μm, 100 μm and 200 μm lengths with 3 μm, 7 μm and 13 μm separations). (**f**) THz transmittances (logarithmic plot) at 0.2–2.0 THz for 305 K (blue lines) and 375 K (red lines). Reproduced from [23], with permission from American Chemical Society 2010.

**Figure 13 nanomaterials-11-00114-f013:**
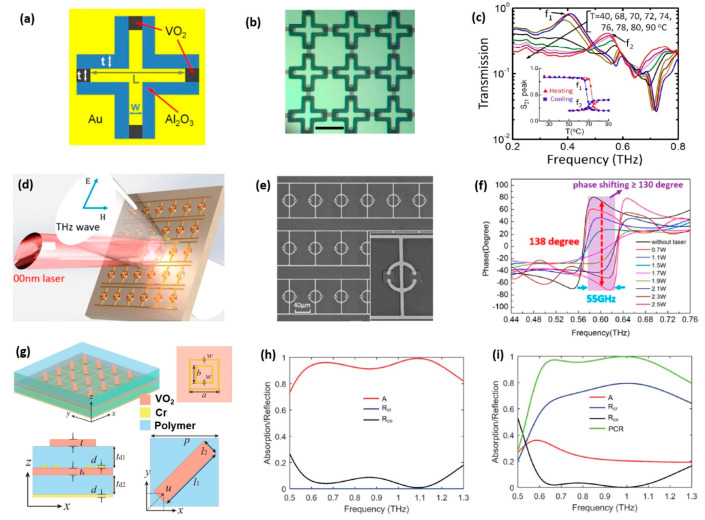
Metal VO_2_ hybrid metamaterials. (**a**) Schematic view of a single loop cross dipole (LCP) unit and (**b**) the optical microscope image (scale bar corresponds to 100 um). (**c**) Measured S_21_ coefficients of the LCP as a function of temperature during the heating process. The inset shows *f*_1_ and *f*_2_ peak positions versus temperature in heating and cooling cycles. Reproduced from [30], with permission from Optical Society of America 2013. (**d**) Schematic of the photoinduced phase converter fabricated by VO_2_ hybrid photoconductive composite structure (PCS) and (**e**) the SEM images of the device. (**f**) The phase diagram of the PCS triggered by different laser fluence. Reproduced from [26], with permission from American Chemical Society 2018. (**g**) Schematic of a VO_2_-integrated THz device with switchable functionalities. The dimensions in the side and top view are *p* = 100 um, *l*_1_ = 110 um, *l*_2_ = 26 um, *t* = *t*_s_ = 1 um, *td*_1_ = 40 um, *td*_2_ = 34 um, *a* = 55 um, *b* = 36 um, *w* = 1 um and *d* = 0.3 um, respectively. (**h**) Simulated absorption, copolarized reflection and cross-polarized reflection at normal incidence when VO_2_ is in the insulating state with σ = 200 S/m. *A*, *R*_co_, *R*_cr_ and PCR represent the absorption, cross-polarized reflection, copolarized transmission and polarization conversion of the device, respectively. (**i**) Simulated absorption, copolarized reflection, cross-polarized reflection, and PCR at normal incidence when VO_2_ is in its fully metallic state with σ = 200,000 S/m. Reproduced from [25], with permission from WILEY-VCH 2018.

**Figure 14 nanomaterials-11-00114-f014:**
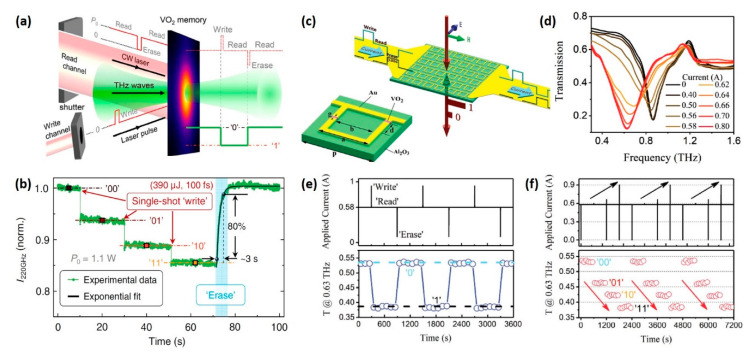
THz memorizer based on the phase transition of VO_2_. (**a**) Schematic of the all-optically driven 2-bit memory device based on VO_2_ film. A continuous-wave laser as a read channel provides a constant bias power P0 to maintain the temperature of VO_2_ film at the percolation threshold of IMT, and shutting off P0 means the “erase” operation. The states of VO_2_ film are written by fs laser (1560 nm) and detected by transmitted THz waves (220 GHz) in real time. (**b**) Time-dependent 2-bit memory effect of VO_2_ written by three fs pulses. The bias pump power is P0 = 1.1 W, the writing pulse fluence is 5.5 mJ/cm^2^, and the pulse duration is 100 fs. The exponential fit (thick black curve) to the data after the “erase” execution demonstrates the thermal equilibrium time of ~3 s, according to the 10–90% criterion. Reproduced from [33], with permission from Optical Society of America 2020. (**c**) Schematic of the electric filed controlled meta-device fabricated by VO_2_ hybrid asymmetric split-ring resonators (VO_2_-ASRR) and a single unit of the VO_2_-ASRR. (**d**) Frequency spectrum as a function of applied current. (**e**) Timing diagram of the binary programming process. Applied current pulses: 1 A (1 s) for “write,” 0 A (2 s) for “erase,” 0.58 A for “read,” (continuous bias). The state of the metadevice was detected by the THz transmission at 0.63 THz. (**f**) Timing diagram of the 2-bit programing process. Applied current pulses: 0–0.58 (“00”)—0.6–0.58 (“01”)—0–0.66–0.58 (“10”)—0–0.9–0.58 A (“11”). Four states denoted as “00,” “01,” ”10” and “11” can be distinguished by the THz transmission amplitude at 0.63 THz. Reproduced from [35], with permission from WILEY-VCH 2018.

**Figure 15 nanomaterials-11-00114-f015:**
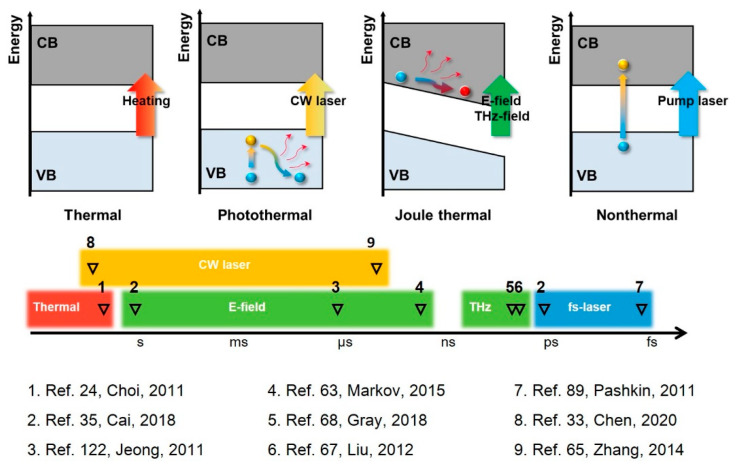
Overview of the available modulation schemes for VO_2_-based optical devices in the THz regime: Modulation approaches based on heating (red arrow), CW laser (yellow arrow), electric field and intense THz field (green arrow) and pump fs laser (blue arrow), with thermal, photo89thermal, Joule thermal and direct photoexcitation (nonthermal) mechanism. The range of response time of the diverse modulation approaches are plotted as red, yellow, green and blue bars with experiment data presented as black triangles. The data are adapted from [24,33,35,63,65,67,68,89,122].

## Data Availability

Data available in a publicly accessible repository.

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
