# Peer review of "Dynamic Manipulation of THz Waves Enabled by Phase-Transition VO2 Thin Film"

_nanomaterials, 2021, doi:10.3390/nano11010114_

Round 1

Reviewer 1 Report

This manuscript reviews all possible applications using the IMT induced by electro-optical method such as photo-driven IMT. Authors studied all research papers using optical method, in particular, THz researchs. The applications are enumerated as phase-change memory, metamaterials, optical switching in a large region. Their analysis's are reasonable. This was well organized and written. I think it is nearly perfect. English is also easy to understand the sentences. I think this can largely help researcher optically studying the phase change materials. I found one error in references; ref. 83 and ref. 84 are same. I hope authors correct the simple error. I recommend this for publication.

Author Response

We appreciate greatly your kind comments. We have fixed the mentioned error and rechecked the reference list. Thanks again for your careful proofreading.

Reviewer 2 Report

The authors presented a review manuscript on the THz device applications based on the metal-insulator transition of VO2 films with the main focus on the modulation of the VO2 conductivity in different metamaterials in the THz region of spectrum. Being impressed by the chosen topic and presentation of basic results, I have some comments and questions, which I believe would help to improve the quality of the manuscript.

In the introduction the authors started directly with VO2 related THz applications, but I would be also happy to see a more motivated introduction in THz topic itself: why and where these applications could be important, what is the status quo of the THz optics/electronics? I believe such extended introduction will attract more readers, which not necessary are dealing with THz business.

I appreciate the presented broad description of VO2 related phenomena in Chapter 2, which I found very good and necessary, but some points and wording have to be more thoroughly formulated. Namely, the phase transition. In the abstract one can see “insulator-metal phase transition”, which in the sense of Landau theory of phase transitions cannot be called “phase transition” as for the IMT one cannot really introduce the order parameter. A more correct would be “insulator-metal transition”. In page 4 (lines 151-153) the authors discussed different manifestations of the IMT in single crystals and films “Unlike the abrupt phase transition in single-domain VO2 crystals, the IMT of VO2 films is continuous and always exhibits an inhomogeneous region where the insulating phase and the metallic phase coexist [83-85]”. Both crystals and films of VO2 display a 1st order phase transition with pronounced hysteresis, i.e. the phase is discontinuous. But pinning of electronic/structural phases on defects in thin films results in a peculiar phase separated behavior, which is not easily to detect in crystals.

Concerning the photoinduced IMT (see 3.2, p. 6, lines 238-241), I would expect stricter definitions, because if the authors assume that “… researchers believe that the modulation phenomenon excited by CW laser is mainly a photo-thermal process”, then the photoinduced IMT is simply does not exist. Even in the case of fs-pulse and seemingly instantaneous IMT at the (fs-ps) time scale, the Fig. 5 reveals a much more complicated nature of the IMT transition actually developing at a much larger time scale, i.e. when the structure changes. This problem is very complex and fundamental and, likely, is not of so drastic importance for the topic of this manuscript. But many authors from the “fs” community often use the formulations of “ultrafast phase transition” neglecting its classic “steady state” complex nature.

The resolution of figures is low and the details of the text and symbols are difficult to see. Please, use the high resolution pictures.

In addition, there are several problems with the text, for example: a) the sentence in p. 9 (lines 338-340) is not correctly written, please fix; b) In the legend to Fig. 9 is written: “The modulation depth along bR-axis (~85%) is dramatically different from that along cR (~15%)”. According to the Fig. 9, it should be other way around, i.e. “bR-axis (15 %) and cR(85%)”; c) there are a lot room to improve the text grammar and style, I would recommend to ask for the help of native English speaker – instead of “decided” in the context of p. 16 (line 642) should be changed to “determined”.

Summarizing, I believe the manuscript can be published in Nanomaterials (MDPI) after authors considered my comments and changed the manuscript accordingly.

Author Response

Response to Reviewer 2 Comments

We appreciate greatly your constructive comments. All the comments have been addressed below, and revisions have been made in the text where appropriate.

Point 1: In the introduction the authors started directly with VO2 related THz applications, but I would be also happy to see a more motivated introduction in THz topic itself: why and where these applications could be important, what is the status quo of the THz optics/electronics? I believe such extended introduction will attract more readers, which not necessary are dealing with THz business.

Response 1: Thanks for your suggestion. We have added a brief introduction to the necessity of THz technology in the Introduction section (see p.2, lines 47-55) as follows:

“More recently, the rapid development of multimedia service is causing explosive demand for high-capacity wireless communications. THz communication technology has become more and more important for the potential of increased bandwidth capacity compared to microwave systems. The manipulation of the transmission properties of THz waves such as amplitude, phase, polarization and spatiotemporal distribution, is based on the modulation effect of THz modulators, which is one of the core devices in THz communication system. Practical applications require THz modulators capable of effectively manipulating the electromagnetic properties of THz waves and dynamically responding to external control signals, which significantly promote the research and application of VO2 in the THz regime.”

Point 2: I appreciate the presented broad description of VO2 related phenomena in Chapter 2, which I found very good and necessary, but some points and wording have to be more thoroughly formulated. Namely, the phase transition. In the abstract one can see “insulator-metal phase transition”, which in the sense of Landau theory of phase transitions cannot be called “phase transition” as for the IMT one cannot really introduce the order parameter. A more correct would be “insulator-metal transition”.

Response 2: Thanks for your professional suggestion. We have replaced the term “insulator-metal phase transition” with “insulator-metal transition” and avoided using the word “phase transition” when we refer to the insulator-metal transition of VO2.

Point 3: In page 4 (lines 151-153) the authors discussed different manifestations of the IMT in single crystals and films “Unlike the abrupt phase transition in single-domain VO2 crystals, the IMT of VO2 films is continuous and always exhibits an inhomogeneous region where the insulating phase and the metallic phase coexist [83-85]”. Both crystals and films of VO2 display a 1st order phase transition with pronounced hysteresis, i.e. the phase is discontinuous. But pinning of electronic/structural phases on defects in thin films results in a peculiar phase separated behavior, which is not easily to detect in crystals.

Response 3: Thanks greatly for your professional suggestion. According to your comments, we have replaced the original misleading descriptions with “But for multi-domain VO2 thin films, the conductivity transition is much more complex due to the dispersion of local phase-transition temperature in different domains” (see p.4 lines 164-165). We think that compared with “IMT” the term “conductivity transition” is more suitable when we refer to the macroscopic conductivity change of the whole VO2 film.  

Point 4: a) Concerning the photoinduced IMT (see 3.2, p. 6, lines 238-241), I would expect stricter definitions, because if the authors assume that “… researchers believe that the modulation phenomenon excited by CW laser is mainly a photo-thermal process”, then the photoinduced IMT is simply does not exist. Even in the case of fs-pulse and seemingly instantaneous IMT at the (fs-ps) time scale, the Fig. 5 reveals a much more complicated nature of the IMT transition actually developing at a much larger time scale, i.e. when the structure changes. b) This problem is very complex and fundamental and, likely, is not of so drastic importance for the topic of this manuscript. c) But many authors from the “fs” community often use the formulations of “ultrafast phase transition” neglecting its classic “steady state” complex nature.

Response 4: According to your constructive suggestions, we have reorganized the contents relating to photo-induced IMT, mainly as follows:

  1. We have deleted the original misleading definitions in Section 3.2 (p.6) that attribute the IMT mechanism to photo-thermal effect or photo-doping effect according to the form of the optical source (CW laser or intense pump pulse). Based on the research by Zhai et al. ( Express 2018, Vol. 26, No. 21), the photo-thermal and photo-doping effects are two competing mechanisms in the photo-induced IMT process and the interaction between them is complex. So we weakened the definition of detailed physical mechanism and only introduced the experimental phenomenon emphasized in different kinds of experiments (see p.6 lines 251-264).           
  2. We want to keep the introduction concerning the ultrafast IMT in Section 3.2 since it is aimed at explaining why only using intense pulse laser can researchers observe ultrafast modulation phenomenon in VO2. In this section we emphasis the dependence of IMT modulation depth on pump laser fluence and sample temperature. We hope these introductions can help to design THz devices based on the ultrafast IMT of VO2.  
  3. According to your professional suggestion, we have checked our manuscript to avoid putting the word “ultrafast” together with “phase transition”.

Point 5: The resolution of figures is low and the details of the text and symbols are difficult to see. Please, use the high resolution pictures.

Response 5: We feel sorry that we did not provide pictures with enough resolution. All the figures have been optimized with special attention on figure resolution and symbol.

Point 6: In addition, there are several problems with the text, for example: a) the sentence in p. 9 (lines 338-340) is not correctly written, please fix; b) In the legend to Fig. 9 is written: “The modulation depth along bR-axis (~85%) is dramatically different from that along cR (~15%)”. According to the Fig. 9, it should be other way around, i.e. “bR-axis (15 %) and cR(85%)”; c) there are a lot room to improve the text grammar and style, I would recommend to ask for the help of native English speaker – instead of “decided” in the context of p. 16 (line 642) should be changed to “determined”.

Response 5: We feel great thanks for pointing out our mistakes. We have fixed the mentioned errors and rechecked our manuscript and improved some text.

  1. The sentence in p. 9 (lines 338-340) has been fixed and is now in p. 9-10 (lines 371-374) in the revised manuscript.
  2. The error in the caption of Figure 9 has been fixed as “the modulation depth along bR-axis (~15%) is dramatically different from that along cR (~85%).”
  3. The word “decided” in p. 16 (line 642) has been be changed to “determined” and is now in p. 17 (lines 656) in the revised manuscript.

Thanks again for your careful proofreading.

Reviewer 3 Report

This is a good review article that provides a comprehensive review of the recent advances in VO2 based materials and their tuning in the THz range. Authors have reviewed both electrical properties of VO2 upload phase transition as well as their implementation in various applications in THz. Apart from some minor typos, this is very well written article and acceptable for publication.

Author Response

We appreciate greatly your kind comments. We have rechecked our manuscript and fixed some spelling and grammar mistakes. Thanks again for your careful proofreading.